# Associations between News Media Coverage of the 11 September Attacks and Depression in Employees of New York City Area Businesses

**DOI:** 10.3390/bs11030029

**Published:** 2021-02-27

**Authors:** Betty Pfefferbaum, Jayme M. Palka, Carol S. North

**Affiliations:** 1Department of Psychiatry and Behavioral Sciences, College of Medicine, University of Oklahoma Health Sciences Center, P.O. Box 26901, WP3217, Oklahoma City, OK 73126-0901, USA; 2Department of Psychiatry, University of Texas Southwestern Medical Center, 6363 Forest Park Road, BL316, Dallas, TX 75390-9070, USA; Jayme.Palka@utsouthwestern.edu; 3Metrocare Services, 1250 Mockingbird Lane, Suite 330, Dallas, TX 75247-4914, USA; Carol.North@utsouthwestern.edu; 4Department of Psychiatry, University of Texas Southwestern Medical Center, 5323 Harry Hines Blvd., Suite NE5.102, Dallas, TX 75390-9070, USA

**Keywords:** depression, depression symptoms, disaster, functional impairment, major depressive disorder, media, news media, 11 September 2001 attacks, terrorism, trauma exposure

## Abstract

Research has examined the association between contact with media coverage of mass trauma events and various psychological outcomes, including depression. Disaster-related depression research is complicated by the relatively high prevalence of the major depressive disorder in general populations even without trauma exposure. The extant research is inconclusive regarding associations between disaster media contact and depression outcomes, in part, because most studies have not distinguished diagnostic and symptomatic outcomes, differentiated postdisaster incidence from prevalence, or considered disaster trauma exposures. This study examined these associations in a volunteer sample of 254 employees of New York City businesses after the 11 September 2001, terrorist attacks. Structured interviews and questionnaires were administered 35 months after the attacks. Poisson and logistic regression analyses revealed that post-9/11 news contact significantly predicted the number of postdisaster persistent/recurrent and incident depressive symptoms in the full sample and in the indirect and unexposed groups. The findings suggest that clinical and public health approaches should be particularly alert to potential adverse postdisaster depression outcomes related to media consumption in disaster trauma-unexposed or indirectly-exposed groups.

## 1. Introduction

Research has examined the association between contact with media coverage of mass trauma events and various psychological outcomes. Posttraumatic stress outcomes—posttraumatic stress disorder (PTSD) and posttraumatic stress symptoms—have been most widely studied [1,2,3], but other outcomes including depression have also been investigated [2,4]. In particular, media coverage of the 11 September) attacks generated extensive research on media contact and posttraumatic stress [1,2] with fewer studies focused on depression [2]. The association between 9/11 media contact and depression was assessed in general populations in the New York City (NYC) area [5] and nationwide in the United States (U.S.) [6]. It was also examined in samples with direct connections to the attacks including U.S. Pentagon employees, some of whom were onsite at the time of the Pentagon attack [7], and undergraduate college students in Ohio who evacuated their university when United Air Flight 93 circled overhead prior to crashing in Pennsylvania [8]. Studies of other disasters examined the association between media contact and depression in general population or community samples [9,10] and in samples of individuals bereaved by disaster [11] and in psychiatric [12] and pain [13] patients. A qualitative review of disaster media research revealed an association between disaster-related television viewing and depression in the relatively few studies examining depression outcomes [2]. A recent meta-analysis found a small statistically significant association between mass trauma media contact and depression [4].

### 1.1. The Importance of Examining Depression

Disaster-related depression research is complicated by the relatively high prevalence of major depressive disorder (MDD) in general populations even without trauma exposure, with a lifetime prevalence of 16.2% [14]. Postdisaster MDD is generally less prevalent than PTSD [15,16]. Examining MDD in directly-exposed survivors of 10 disasters using a consistent rigorous diagnostic methodology, North and colleagues [17] found a postdisaster MDD incidence of 6% and disaster-related PTSD in 16%. Interestingly, evidence suggests that MDD was especially prevalent relative to PTSD after the 9/11 attacks [18,19,20]. Unlike PTSD, criteria for MDD do not require exposure to trauma, but the development of MDD and depression symptoms may be influenced by the individual’s specific disaster exposure [18,20,21]. Because of the relatively high prevalence of depression in both general and disaster-affected populations, clinicians should consider depression in assessing and managing psychiatric and medical patients after disasters, and public mental health practitioners should consider the potential implications of postdisaster MDD for the general public.

### 1.2. The Current Study

A review of disaster media studies [2] and a recent meta-analysis of the association of media contact and depression [4] identified a limited number of studies of depression relative to studies of posttraumatic stress outcomes and a paucity of studies that measured diagnostic outcomes. The meta-analysis failed to find differences related to participants’ disaster exposure [4]. Additionally, few studies have distinguished prevalence and incidence in outcomes. This precludes the differentiation of cases that were new after the disaster, and possibly caused by it, from other postdisaster psychopathology with onset prior to, and thus unrelated to, the disaster. The current study of employees of NYC area businesses after the 9/11 World Trade Center (WTC) attacks addressed these considerations. The study examined the association of media contact with pre- and postdisaster depression symptoms, MDD, and functional impairment (using full DSM-IV diagnostic criteria) [22]; considered pre- and postdisaster time frames to assess both postdisaster prevalence and incidence of the depression variables; and examined these variables within disaster trauma exposure groups (delineated by DSM-IV PTSD criteria). Relationships of these variables with disaster-related media consumption may have important implications for clinical and public mental health practice.

## 2. Materials and Methods

In the original study from which data for this analysis were collected, a volunteer sample of 379 employees of eight businesses affected by the 9/11 attacks in NYC, including 176 employees of three businesses located in the WTC towers, was recruited through information about the study distributed at the workplace, at a mean of 35 months (range, 27–52 months) after the attacks. Extensive detail about the sample methods and determination of 9/11 trauma exposures is available in previous publications [20,23,24]. The Institutional Review Boards of the collaborating academic institutions for this study approved the project. All participants provided written informed consent.

### 2.1. Assessments

Structured interviews were administered by mental health professionals formally trained on these interviews. Pre- and postdisaster psychiatric disorders were assessed with the Diagnostic Interview Schedule (DIS) for DSM-IV (DIS-IV) [25] by obtaining onset and recency data relative to the date of the 9/11 attacks for diagnosed psychiatric disorders. A DSM-IV diagnosis of MDD required ≥1 episodes representing a change from the individual’s usual state lasting at least two weeks with ≥5 of 9 symptoms including low mood and/or loss of interest/pleasure, not explained by effects of a substance or general medical condition, and causing clinically significant distress or impairment in social, occupational, or other areas of functioning. The main outcome variables were the depressive episode symptom count, MDD diagnosis (yes/no), and associated functional impairment (yes/no) in pre- and postdisaster time frames. Postdisaster prevalence (all cases after the disaster) was distinguished from incidence (only postdisaster cases that were never present before the disaster).

Detailed information about participants’ disaster experiences including trauma exposure was collected with the Disaster Supplement [26]. Exposures were categorized per DSM-IV PTSD criteria as direct, witnessed, and indirect through the learning of the trauma exposure of a close family member or close friend [22]. The exposure categories were not hierarchical or mutually exclusive, with exposure groups assigned regardless of other exposure types, allowing multiple exposure types.

The self-report Disaster Supplement Questionnaire (DSQ) [27] collected detailed information about contact with news media coverage of the 9/11 attacks and subjective perceptions of it. Only 254 participants (67%) completed the DSQ, constituting the sample for this analysis; the only difference for the DSQ non-completion sample was higher disaster-related PTSD prevalence (20% vs. 10%; χ^2^ = 6.52, df = 1, *p* = 0.011), potentially biasing the study sample toward psychological resilience. Participants estimated the number of hours a day they spent “obtaining news from all sources” in the month before the 9/11 attacks and the week after the attacks.

### 2.2. Data Analysis

Data analysis was conducted using SAS 9.4 (SAS Institute, Cary, NC, USA). Prior to conducting statistical analyses, descriptive findings were derived with the results presented as frequencies and proportions for categorical variables and as means and standard deviations (SD) for numerical variables. Categorical comparisons used two-sided χ^2^ analyses, substituting Fisher’s exact texts for expected cell sizes <5. Predisaster depression variables (i.e., number of predisaster symptoms, MDD diagnosis, and functional impairment) were analyzed as both pre-9/11 and post-9/11 predictors of the number of hours of media contact using the linear regression procedures in SAS (PROC REG). Outcomes representing numbers of persistent/recurrent and incident depressive symptoms—treated as count variables—were analyzed using the generalized linear model procedures in SAS (PROC GENMOD) specified for a Poisson distribution and associated variance function. Postdisaster persistent/recurrent and incident dichotomous outcomes (i.e., MDD diagnosis and functional impairment) were analyzed using binary logistic regression analyses. All analyses were performed separately in subsamples stratified by type of trauma exposure. Statistical significance was set at α ≤ 0.05 for single comparisons. Corrections for multiple comparisons were made within each time frame for either prevalence or incidence (15 comparisons across depression symptoms, MDD diagnosis, and functional impairment in 5 exposure groups corrected separately in pre-9/11 and post-9/11 time frames) using the Bonferroni method by dividing the usual α level of 0.05 by 15 (α ≤ 0.003).

## 3. Results

The sample (*n* = 254) was approximately one-half (45%, *n* = 115) male, median age 45 years, more than two-thirds (69%, *n* = 179) white, about two-thirds (66%, *n* = 168) college-educated, and about one-half (51%, *n* = 129) currently married. More detail regarding demographics and other characteristics of the sample is available in a previous publications [24]. Nearly one-half of the sample (41%, *n* = 105) had a 9/11 trauma exposure. Approximately one-fourth (22%, *n* = 57) were directly exposed, including 36 individuals in the WTC towers and another 21 immediately outside the towers during the attacks; 15% (*n* = 37) were indirectly exposed.

The mean (SD) number of depression symptoms in the full sample was 2.7 (3.2) for predisaster symptoms, 1.6 (2.2) for postdisaster persistent/recurrent symptoms, and 0.7 (1.7) for incident symptoms. The proportions of the full sample meeting MDD criteria were 30% (*n* = 76) for predisaster MDD, 19% (*n* = 42) for postdisaster persistent/recurrent MDD, and 10% (*n* = 26) for incident MDD. Impairment from depression symptoms in the full sample was reported by 19% (*n* = 47) for predisaster impairment, 8% (*n* = 20) for postdisaster persistent/recurrent impairment, and 6% (*n* = 14) for incident impairment.

The mean number of hours a day of news media contact was significantly higher in the week after the attacks (mean = 5.7 h; SD = 4.8) than in the month before the attacks (mean = 2.1 h; SD = 2.1; t(225) = 12.67, *p* < 0.001).

### Predictive Ability of Media Variables with Respect to Outcomes

No demographic variables were associated with the amount of news media contact in the month before or the week after the disaster. Table 1 shows results from analyses in which hours of pre-9/11 and post-9/11 media contact were regressed onto predisaster depression variables (number of depression symptoms, MDD, and functional impairment). Neither of the predisaster depression variables nor the predisaster functional impairment variable significantly predicted pre-9/11 or post-9/11 news contact.

Table 2 shows results from analyses in which postdisaster depression variables were regressed on media contact. Postdisaster media contact was a positive predictor of the number of postdisaster persistent/recurrent depression symptoms in the full sample (β = 0.05, *p* < 0.001), the indirect exposure group (β = 0.07, *p* < 0.001), and the group with no exposure (β = 0.06, *p* < 0.001). Media contact was not a significant predictor of postdisaster persistent/recurrent MDD or functional impairment from depression symptoms.

Table 3 shows results from analyses in which incident depression variables were regressed onto media contact. Post-9/11 media contact was a significant positive predictor of the number of incident depression symptoms in the full sample (β = 0.05, *p* < 0.001) and in the groups with indirect exposure (β = 0.07, *p* < 0.001) or no exposure (β = 0.09, *p* < 0.001). Media contact was not a statistically significant predictor of incident MDD or incident functional impairment from depression symptoms.

## 4. Discussion

Most of the extant disaster and terrorism media research examining depression has assessed symptoms rather than diagnoses using postdisaster prevalence measures. These studies have identified symptoms or disorders present any time post disaster regardless of whether they were present before the disaster, thus including relapsed and continuing depression symptoms or disorders as well as new (incident) symptoms or disorders. Only MDD and depression symptom onset after the disaster, and not pre-existing symptoms and disorders, can represent activation by the disaster. Additionally, much of the extant disaster media research on depression has not assessed outcomes in distinct exposure groups. Failure to distinguish between diagnostic and symptomatic outcomes, to differentiate postdisaster incidence from prevalence, and to consider trauma exposure may yield misleading findings and interpretations that have potentially important clinical and public health implications. The current analysis was designed to address these shortcomings by assessing both symptoms and diagnoses and by examining associations of pre-9/11 and post-9/11 media contact with pre- and postdisaster prevalence and incidence of depression outcomes in distinct 9/11 trauma exposure groups.

### 4.1. Associations between Media and Depression Outcome Variables

In the current study, post-9/11 media contact predicted higher postdisaster persistent/recurrent and incident depression symptom counts in the full sample and in the indirect and unexposed groups. There was no association between post-9/11 media contact and either postdisaster persistent/recurrent or incident MDD in any of the exposure groups. The prediction of the number of postdisaster persistent/recurrent and incident depression symptoms by media contact in the full sample—accounted for by those with indirect and no exposure—is supported by a recent meta-analysis which found a small effect of media coverage on postdisaster depression symptoms [4]. This suggests that individuals who are not directly exposed to disaster trauma—both those whose prior symptoms persist or recur and those who develop new depression symptoms after a disaster—may seek disaster-related media coverage or that media contact may increase depression symptoms.

A recent meta-analysis of the association between media contact and depression [4] identified only three adult studies that measured clinical depression outcomes [5,7,28]. None of these studies distinguished postdisaster prevalence and incidence, and only one used rigorous diagnostic assessment that considered all diagnostic criteria [28]. Similar to the current findings, that study found no association between Oklahoma City bombing media contact and MDD in a sample of highly-exposed survivors [28]. It is not surprising that the current study failed to find associations between media contact and either postdisaster persistent/recurrent or incident MDD in any of the exposure groups. Although media contact may create distress, one would not expect it to generate depressive illness.

### 4.2. Considerations Related to Disaster Trauma Exposure

Consideration of disaster trauma exposure is central to understanding the relative and distinct roles of media contacts in postdisaster outcomes. Findings that post-9/11 media contact predicted postdisaster depression outcomes in indirectly-exposed and unexposed individuals and not in those with direct or witnessed exposures suggest that postdisaster depression outcomes in directly-exposed individuals are related to their in-person disaster experiences which likely outweigh the effects of media contact. In contrast, in those with indirect or no exposure, the influence of media contact may prevail, generating a variety of emotional reactions including depression symptoms [29].

Concerns that may arise when different exposure groups are not examined separately are illustrated by contrasting the current study with a 9/11 media study by Ahern and colleagues [5] which found an association between repeated viewing of specific televised 9/11 images and depression symptoms in those “directly affected” by the attacks but not in others (p. 296). Instead of separating trauma-exposed (direct, witnessed, and indirect) from otherwise adversely-affected trauma-unexposed groups, different types of disaster trauma exposure (e.g., witnessed in person, friend or family member killed) and experiences that did not represent trauma but were related to the disaster (e.g., lost possessions, displacement from home, participation in rescue efforts, 9/11-related job loss) were included together in the “directly affected” category in that study. Thus, it is unclear the extent to which the reported outcomes arose from different types of disaster trauma exposure or from other non-trauma disaster-related experiences. The careful delineation in the current study of disaster trauma exposure groups from one another and from groups that were not exposed to trauma but were affected as employees in the geographical area allowed comparison of depression outcomes within these different groups.

There is a paucity of literature explicating the mechanism(s) by which media contact is associated with emotional outcomes [29]. Media coverage, especially provocative coverage, may generate a subjective perception and appraisal of threat, fear, and arousal in consumers [29]. Characteristics of media contact (e.g., media form, duration, frequency, context) and coverage (e.g., content, visual and auditory quality) may contribute to the perception and appraisal of threat [29]. For those with direct or witnessed exposure, threat perception and appraisal may rekindle the arousal associated with the direct experience leading to posttraumatic stress. The results of the current study suggest that these appraisals may not lead to depression. For those with indirect exposure, the mechanism for the development of depression may be through the experiences or the loss of a close family member or friend. The findings of an associations of depression symptoms with indirect but not direct or witnessed exposure are consistent with the results of a study of 11 disasters [16] in which only indirect exposure to disaster trauma and no other exposure types predicted MDD. An analysis of data from 10 of these disasters found that indirect exposure did not significantly add to the prediction of PTSD independent of the effects of other predictive variables in the model [17]. The findings of these two analyses together suggest potentially distinct pathways for posttraumatic stress and depression outcomes after disasters, with direct trauma exposure leading to posttraumatic stress and indirect exposure predominantly leading to depression [16]. The results of the current study suggest that the role of disaster-related media contact in the development of postdisaster depression symptoms may be particularly pronounced and problematic for individuals indirectly exposed through close relationships with trauma-exposed survivors and for those not exposed to disaster trauma. Future research should examine the roles of indirect exposure, loss, and other disaster experiences (e.g., experiences related to community destruction) in explaining the underlying mechanisms for these associations.

### 4.3. Outcome Prevalence and Incidence

The extant research on disaster media effects has not addressed the distinction between prevalence and incidence in outcomes. The results of the current study suggest that media contact predicts both persistent/recurrent and incident depression symptoms only in indirectly-exposed and unexposed groups and that it does not predict persistent/recurrent or incident MDD in any exposure group. While the study did not reveal differences in post-9/11 media contact as a predictor of the prevalence and incidence of depression symptoms, MDD, or functional impairment outcomes across exposure groups, the investigation constitutes an important contribution to encourage future work examining both prevalence and incidence.

### 4.4. The Role of Predisaster Depression

In this study, pre-9/11 and post-9/11 media contact were not predicted by predisaster depression symptom count, MDD, or functional impairment. Few studies have examined the influence of predisaster psychiatric disorders on disaster-related media contact, and the results have been inconclusive. In their Boston Marathon bombing study of a nationally-representative sample (about 10% directly or indirectly exposed to the bombing and about 9% directly or indirectly subjected to postdisaster lockdown), Jones and colleagues [30] found that pre-bombing psychiatric history was not associated with consumption of bombing-related media coverage through either traditional or newer media forms. Another study of a nationally representative sample, however, reported a small but significant association between Boston Marathon bombing-related media contact and pre-existing depressive or anxiety disorders [31]. Clearly, research is needed to determine how predisaster psychiatric disorders may influence postdisaster media consumption and its effects.

### 4.5. Strengths and Limitations

Although the current study was limited by its non-representative volunteer sample, a major strength was its sample recruitment from businesses affected by the 9/11 attacks rather than from the general community, permitting assessment of large numbers of individuals with various types of disaster exposure from businesses located at varying proximity to the WTC site. Methodological rigor in this study included the use of structured interviews for a full assessment of diagnostic criteria to specify pre- and postdisaster prevalence and incidence of depression symptoms, MDD, and functional impairment. A potential source of bias was the higher prevalence of disaster-related PTSD in one-third of the original sample not completing the media items and thus excluded from the current analysis.

Another limitation involved changes in the media landscape since this study was conducted, thus raising concern about the relevance of media consumption behaviors occurring so long ago, as media has changed markedly in the last two decades. Notably, while the internet was in use at the time of 9/11, its popularity has increased markedly over the last 20 years. In addition, the introduction of social media over this time period has likely altered behavior with respect to the use of various media forms. The media questions reported in this study addressed “all sources” of news coverage which would include all media forms used by this population at that point in time. The inclusive global measurement of media, referencing news from all sources rather than specific media forms (e.g., newspaper, television, radio, internet) separately, may have influenced the results. Contact with television coverage, the most widely-studied form of disaster media, has been found to be associated with posttraumatic stress [1,2,3]. Less evidence clearly links contact with other forms of media coverage and psychological outcomes including depression [2]. Unexamined factors related to media coverage (e.g., its content or specific characteristics) or media behaviors (e.g., motivations for media contact, avoidance or discontinuation of media contact) may have influenced the results as well and should be examined in future research.

This study’s timing nearly three years after the disaster was another potential limitation. For example, participants might not have remembered the exact number of hours they consumed media coverage before and after 9/11, but they were likely to appreciate differences in their media behavior in the aftermath of 9/11, thus providing important information for a subjective comparison regarding their relative media use in the two time periods studied.

There is a dearth of literature on recall bias in psychological reactions to media coverage. With respect to memory for historic events, memory for experiences related to momentous events such as the 9/11 attacks may be relatively consistent. For example, a study of a nationally representative sample conducted within a month after 9/11 with follow-up assessments in September 2002 and 2003 revealed that consistency of memory for questions related to their 9/11 experiences was relatively high [32]. At least 75% of participants responded consistently to questions related to their 9/11 experiences, and nearly 50% consistently remembered all of the items in 2002 with little decline in 2003 [32].

Another line of research addressing memory for emotion and the potential for emotions to influence memory suggests that memory for prior emotion is largely accurate, but that people both over- and underestimate the intensity of their past emotions relative to an earlier report [33]. Intense emotional experiences are likely to be overestimated because people tend to focus on the most emotionally salient aspect of the experience [33]. Negative experiences, including exposure to traumatic events, generally have a greater impact on individuals than positive experiences [34]. While people tend to overestimate both positive and negative emotions on retrospective recall, their overestimation of negative emotion is likely to be more prominent [34,35].

Relevant to the current study, contemporary emotional states—including depression [35], psychological distress [36], and PTSD [37]—may influence the accuracy of recollection. For example, relative to those who are not depressed, individuals with a depressed mood tend to exhibit more negative than positive memories and to remember more negative emotions [35]. A study of retrospective recall of psychological distress found that university students who were not exposed to trauma overestimated the psychological distress they had reported at the time of an earlier psychoeducational evaluation, but recall bias was not large enough to result in an upward or downward change in the classification of their distress as clinically elevated or non-pathological, respectively [36]. Further, there was no difference between original and retrospective reports of either distress scores or the classification of distress in participants with clinically-elevated scores at the time of the earlier psychosocial evaluation [36]. These findings provide support for the acceptable accuracy of retrospective recall of psychological distress in some non-traumatized populations, but the literature examining recall bias related to trauma experiences is limited. In their study of acute stress disorder (ASD) symptoms, Harvey and Bryant [37] found that the majority of victims of motor vehicle accidents accurately recalled most of the ASD symptom clusters that they had reported in the acute post-event phase, but 75% incorrectly recalled at least one symptom cluster. Current PTSD symptoms two years after the accident influenced the accuracy of recall—posttraumatic stress symptoms were positively associated with errors of addition while low symptom levels were associated with errors of omission [37]. A study of Oklahoma City bombing survivors revealed that the amount of time survivors reportedly spent consuming 9/11 media coverage in the first week after the attacks was independent of the time elapsed before assessment, suggesting that retrospective recall may not create an important source of bias [38]. Thus, while supporting concerns about the accuracy of retrospective reporting, the extant research suggests that the extent of bias may vary across populations and contexts.

Finally, this cross-sectional study examining associations of media contact with pre- and postdisaster depression symptoms, MDD, and functional impairment did not establish causal connections. Disaster media contact may stimulate and/or intensify depression in some, but distressed individuals may use media coverage to seek information or to cope [2,3,29] or they may avoid media coverage to cope or as part of posttraumatic avoidance.

Despite these limitations, the current article addresses important gaps in the literature by examining issues that have not been clarified in research conducted over the last two decades. These issues include attention to the prevalence and incidence of depression outcomes in both pre- and postdisaster time frames within specific exposure groups. The findings reveal that post-9/11 media contact predicts the number of depression symptoms in some exposure groups and indicates population groups that should be the focus of clinical and public health focus.

## 5. Conclusions

This comprehensive analysis of a sample of adult employees of NYC area businesses after the 9/11 attacks extends existing research first by assessing both depression symptoms and diagnosis; second by distinguishing incidence, which most closely captures disaster-related findings, from postdisaster prevalence of depression outcomes; and third by considering the role of 9/11-related media contact in these outcomes in different exposure groups. The findings suggest that postdisaster media contact does not increase the postdisaster prevalence or incidence of MDD and that it predicts new depression symptoms only in those with indirect or no disaster trauma exposure.

The results have implications for addressing disaster-related media coverage in clinical and public mental health practice. The findings suggest that clinicians may want to be particularly alert to potential adverse postdisaster depression outcomes related to media consumption in disaster trauma-unexposed or indirectly-exposed groups. Likewise, public mental health media-focused messaging and interventions should concentrate on these individuals, who may be inclined to seek media contact, and who may benefit from limiting media contact post disaster. Given the relatively large number of indirectly- and unexposed individuals in large-scale major disasters like 9/11, clinical and public health interventions focused on these populations have the potential to benefit many people and to promote recovery across the entire disaster community.

## Figures and Tables

**Table 1 behavsci-11-00029-t001:** Results from linear regression analyses for predisaster depression variables.

	Predisaster Lifetime Prevalence					
	Number of Depression Symptoms		MDD		Functional Impairment from Depression Symptoms	
	*Β*	*p*	*β*	*p*	*β*	*p*
Full sample (*n* = 254)						
Hours/day news contact						
Month pre 9/11	0.01	0.768	−0.01	0.966	0.21	0.550
Week post 9/11	0.19	0.048	0.82	0.230	1.38	0.088
Direct exposure (*n* = 57)						
Hours/day news contact						
Month pre 9/11	−0.01	0.902	−0.35	0.568	−0.25	0.790
Week post 9/11	−0.08	0.795	−1.57	0.400	−0.83	0.795
Witnessed exposure (*n* = 67)						
Hours/day news contact						
Month pre 9/11	−0.03	0.818	−0.59	0.426	−0.12	0.898
Week post 9/11	−0.33	0.133	−2.45	0.098	−2.32	0.215
Indirect exposure (*n* = 37)						
Hours/day news contact						
Month pre 9/11	−0.06	0.708	−0.41	0.691	0.10	0.938
Week post 9/11	0.25	0.367	0.89	0.621	0.06	0.977
No exposure (*n* = 149)						
Hours/day news contact						
Month pre 9/11	0.03	0.538	0.27	0.473	0.33	0.426
Week post 9/11	0.33	0.005	1.90	0.024	2.54	0.007

Note: Bonferroni correction for multiple comparisons in this table defined statistical significance as α ≤ 0.003.

**Table 2 behavsci-11-00029-t002:** Results from poisson and logistic regression analyses for postdisaster depression variables.

	Postdisaster Persistence/Recurrence					
	Number of Depression Symptoms		MDD		Functional Impairment from Depression Symptoms	
	*Β*	*p*	*β*	*p*	*β*	*p*
Full sample (*n* = 254)						
Hours/day news contact						
Month pre 9/11	0.02	0.363	0.03	0.738	−0.07	0.496
Week post 9/11	0.05	<0.001	−0.05	0.161	−0.10	0.006
Direct exposure (*n* = 57)						
Hours/day news contact						
Month pre 9/11	0.03	0.648	0.15	0.633	−0.01	0.987
Week post 9/11	0.01	0.789	0.06	0.536	−0.02	0.844
Witnessed exposure (*n* = 67)						
Hours/day news contact						
Month pre 9/11	0.01	0.007	0.11	0.585	−0.08	0.704
Week post 9/11	−0.01	0.473	0.09	0.311	0.08	0.682
Indirect exposure (*n* = 37)						
Hours/day news contact						
Month pre 9/11	−0.07	0.348	1.03	0.347	7.32	0.838
Week post 9/11	0.07	<0.001	−0.15	0.106	−0.15	0.185
No exposure (*n* = 149)						
Hours/day news contact						
Month pre 9/11	0.04	0.093	−0.04	0.699	−0.09	0.393
Week post 9/11	0.06	<0.001	−0.08	0.074	−0.13	0.004

Note: Bonferroni correction for multiple comparisons in this table defined statistical significance as α < 0.003.

**Table 3 behavsci-11-00029-t003:** Results from Poisson and logistic regression analyses for incident depression variables.

	Incidence					
	Number of Depression Symptoms		MDD		Functional Impairment from Depression Symptoms	
	*Β*	*p*	*β*	*p*	*β*	*p*
Full sample (*n* = 254)						
Hours/day news contact						
Month pre 9/11	−0.20	0.010	0.22	0.281	0.40	0.296
Week post 9/11	0.05	<0.001	−0.05	0.218	0.01	0.917
Direct exposure (*n* = 57)						
Hours/day news contact						
Month pre 9/11	−0.25	0.031	0.35	0.308	0.16	0.649
Week post 9/11	−0.01	0.631	0.02	0.729	0.11	0.305
Witnessed exposure (*n* = 67)						
Hours/day news contact						
Month pre 9/11	−0.31	0.041	0.45	0.315	0.65	0.344
Week post 9/11	−0.02	0.521	0.03	0.686	0.07	0.471
Indirect exposure (*n* = 37)						
Hours/day news contact						
Month pre 9/11	−0.17	0.175	0.18	0.597	0.11	0.781
Week post 9/11	0.07	<0.001	−0.09	0.259	−0.05	0.618
No exposure (*n* = 149)						
Hours/day news contact						
Month pre 9/11	−0.08	0.462	0.06	0.825	7.29	0.851
Week post 9/11	0.09	<0.001	−0.10	0.067	−0.08	0.345

Note: Bonferroni correction for multiple comparisons in this table defined statistical significance as α ≤ 0.003.

## Data Availability

No new data were created or analyzed in this study. Data sharing is not applicable to this article.

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
