# Peer review of "Associations between News Media Coverage of the 11 September Attacks and Depression in Employees of New York City Area Businesses"

_behavsci, 2021, doi:10.3390/bs11030029_

Round 1
Reviewer 1 Report
I have several remarks and problems with this study.
First, this paper will be published somewhere in 2021. This is around 16 years after the study was conducted. Most journals would not even accept survey data of more than 5 years after they are performed.
Second, there is a question whether the data collected after a mean of 35 months (which is not mentioned as such btw in the abstract) is accurate. In the discussion you mention a couple of studies to prove that this would be the case. And I disagree with that.
Example, you state the article "Memory for acute stress disorder symptoms" by Harvey and Bryant. You state that it shows that recall is accurate, "with slight discrepancies". However, when one reads the article, one would draw a different conclusion. In fact, you only need to read the last sentence of the abstract of the article to what I mean.
The other study does support your argument. However, as the author themselves state, this is done with a young population, which is not the case in your study.
The other study you mention, on whether Oklahoma city bomber survivors and media watching 9/11, is done after 9/11. So, you have evidence that people remember how much they watched the biggest event of the decade - and I believe that might be accurate. You have however not proven that they remember how much they watched BEFORE 9/11.
It means that at best, it is unclear whether retrospective studies are accurate.
Third, I do not believe your chosen analyses are accurate. You use lineair models on a sample that is 1, probably not lineair nor normally distributed (as it is a volunteer sample this is very unlikely), 2, you divide that sample into subcategories, where one group for example has an n of 37. It is highly unlikely that you fulfill the assumptions for the tests you have used. Furthermore, you're shooting yourself in the foot by doing this: you should use non-parametric methods, as they have more power with such samples than lineair models - in fact, you might find more significant associations.
Additionally, you are using count data, as you are counting symptoms, no? If so, please check with statistician, as count-data often requires other forms of analysis than non-count data. I'm no expert on count-data however, so I might be mistaken on this aspect.
Fourth, I'm at loss what this study actually contributes to the field. Obviously, it has problems with causation - I don't even mind that, because cross-sectional studies still contribute important results. My problem is with the type of association this study wants to prove. 35 months later, and then you want to prove an association between media usage and depression, before and after 9/11 - okay, perhaps I would still follow that, IF the method was okay. Which it isn't. You have a volunteer sample, in a study on one the most important events of that decade - naturally there is going to be bias. And more importantly, there will be a HUGE bias on how much they watched before 9/11.
Finally, it remains unclear to me how the mechanism of media-watching and mental health would work. Aren't there a hundred mediators or moderators between those two?
Small remark:
- I would insert a dummy table, where you give the characteristics of the sample.
Overall, I think the study is, honestly, not strong enough and incredibly dated. It has issues of bias, and feels like a paper made only because you wanted to write a paper with some data you had left lying around (even ignoring the fact that you are only publishing it now). It has statistical issues, and issues with the interpretation of the results.
Author Response
Comments to All Reviewers:
Thank you for the thoughtful review and for the opportunity to submit a revised version of our manuscript.
We modified the Abstract to reflect the changes in results after conducting a new analysis of the data (lines 26-33).
We simplified the text as much as possible making changes in wording (to clarify the content) and in punctuation (to enhance the flow). We did not track all of these non-substantive changes because it would be distracting to the reader. Substantive changes are identified in red font in the revised manuscript.
We added a recent publication that helps provide justification for the paper in section 1 Introduction (lines 39-57) and in section 1.2 The Current Study under Introduction (lines 74-90).
We edited and moved the original section on Clinical and Public Health Implications to Section 5 Conclusions (lines 401-411).
Response to Reviewer 1:
With respect to the concern about publishing this paper years after the data were collected, we argue that the findings add important knowledge to the field addressing issues that have not been reported heretofore. We added a discussion of this in section 4.5 Strengths and Limitations in the Discussion (lines 319-325) acknowledging the concern and describing changes in the media landscape since this study was conducted. In the revised text, we note that the increased popularity of the internet and the introduction of social media over this time have likely altered behavior with respect to the use of various media forms. The media questions reported in the paper addressed the use of and reactions to “all sources” of news coverage including internet coverage. The original draft of this paper addressed limitations related to the use of this global measurement (now lines 325-333). Despite these limitations, the current paper addresses important gaps in the literature by examining issues that have not been clarified in research conducted over the last two decades. These issues include attention to the prevalence and incidence of both depression symptoms and major depressive disorder and functional impairment in both pre- and postdisaster time frames across exposure groups (lines 384-389).
We concur that recall bias is a major concern and thus identified this as a limitation of the study. We greatly appreciate the reviewer for challenging us on this point as it led us to conduct a more comprehensive review the literature on recall bias and to modify our text to reflect a more sophisticated understanding of the issue. We were unable to identify any literature on memory related to media consumption and thus could not include this in the paper. We recognize that the items querying the number of hours of media contact supposed more precise answers than is likely reasonable especially for pre-9/11 behavior. It might have been better to query a more subjective response but we also think that asking respondents to provide information on their hours of pre- and post-9/11 media contact allowed them to provide their best assessment of their relative media use in the two time periods studied (lines 334-338). We enhanced the discussion of recall bias in section 4.5 Strengths and Limitations of the Discussion (lines 339-378).
We conducted a reanalysis of the data using 1) logistic regression analysis (a nonparametric procedure that does not assume the outcome variable to be normally distributed) for dichotomous x continuous variable comparisons and 2) generalized linear model procedures (PROC GENMOD in SAS) with a non-normality statement specifying a Poisson regression for continuous x continuous variables, which are appropriate for symptom count variables. The revised text on data analysis is presented in section 2.2 Data Analysis under Materials and Methods (lines 128-145). The reanalysis generated changes in the results which are presented in section 3 Results (lines 156-194).
The reviewer questioned the contribution this paper makes to the field. We are intimately familiar with the disaster media research and have made major contributions to the literature in this area including reports of numerous empirical studies. The first author and her team published a review of disaster media effects in 2014 and more recently conducted and published meta-analytic studies of media contact with posttraumatic stress (Pfefferbaum et al 2019), depression (Pfefferbaum et al 2021), and anxiety (Pfefferbaum et al 2021). We did not cite the meta-analysis examining depression outcomes (Pfefferbaum et al 2021) in the original submission of the current manuscript because the meta-analysis paper had not been published, but we were fully aware of the findings. The earlier meta-analysis publications identified issues that warrant attention in future research. In fact, the results of the depression meta-analysis prompted the analysis of the 9/11 data and the current report. Noteworthy, in the recent meta-analysis of depression outcomes were findings of a limited number of studies of depression in adult samples relative to studies of posttraumatic stress outcomes, a paucity of studies that measured clinical outcomes using diagnostic assessment, and the failure to find differences related to participants’ event exposure (Pfefferbaum et al 2021). Thus, the current study focused on these issues and addressed other neglected issues related to the time frame of assessment relative to the event and the distinction between prevalence and incidence. In fact, we chose to report the findings of this study because we think that despite the limitations of the volunteer sample, recall bias, and cross-sectional design, the paper makes an important contribution to the field. We provided the rationale for reporting the results in section 1.2 The Current Study in the Introduction (lines 74-90), in section 4.5 Strengths and Limitations of the Discussion (lines 384-389), and in section 5 Conclusions (lines 401-411).
The reviewer questioned the mechanism for the association between media contact and mental health outcomes and raised concern about mediators and moderators. The mechanism for the relationship between media contact and psychological outcomes has not been subjected to empirical study. We agree that there are a number of potential mediators and moderators between these variables some of which were examined in published meta-analyses of posttraumatic stress (Pfefferbaum et al 2019) and depression and anxiety (Pfefferbaum et al 2021) outcomes. We present information about the mechanisms in section 4.2 Considerations Related to Disaster Trauma Exposure in the Discussion (lines 262-284).
With respect to the inclusion of a dummy table describing characteristics of the sample, this table is published in a prior article (Zettl et al 2020). To avoid possible copyright violations, the need to obtain permission from the publisher of the cited manuscript, and likely payment of a fee for using the table, we instead summarized the relevant findings with the appropriate reference in text in the first paragraph of section 3 Results (lines 149-155).

Reviewer 2 Report
Author(s) can give greater attention to the heterogeneity of the unexposed cohort. Future research can try to differentiate those who may have predilections to associate with tragedy and their likelihood to develop future trauma symptoms through media exposures. The unexposed exposed differentiation is interesting and more speculation can be given to the reasons why this was found to occur.
Author Response
Comments to All Reviewers:
Thank you for the thoughtful review and for the opportunity to submit a revised version of our manuscript.
We modified the Abstract to reflect the changes in results after conducting a new analysis of the data (lines 26-33).
We simplified the text as much as possible making changes in wording (to clarify the content) and in punctuation (to enhance the flow). We did not track all of these non-substantive changes because it would be distracting to the reader. Substantive changes are identified in red font in the revised manuscript.
We added a recent publication that helps provide justification for the paper in section 1 Introduction (lines 39-57) and in section 1.2 The Current Study under Introduction (lines 74-90).
We edited and moved the original section on Clinical and Public Health Implications to Section 5 Conclusions (lines 401-411).
Response to Reviewer 2:
We recognize that there may be some heterogeneity in the unexposed cohort, but the central characteristic of interest in this group is that it was unexposed to disaster trauma, which is the point of examining the findings in this group. To look into any potential effects of this heterogeneity on the findings, we compared the demographic characteristics of this group with the exposed groups, but there were no significant differences, thus arguing against further pursuit of addition of adjustments of these characteristics to the analyses of this group. Importantly, variables in this group were not directly compared to variables in the other exposure groups, further suggesting such adjustments are not needed for the analyses presented in this study.
We provide an enhanced discussion of the differences between the exposed and unexposed groups in section 4.2 Considerations Related to Disaster Trauma Exposure in the Discussion (lines 238-284).

Reviewer 3 Report
The authors mention the relationship between the length of news contact time and depression induced by 9.11 attacks in employees of NYC. Post disaster media contact is related to post disaster depression only with indirect and no exposure to 9.11 attacks, but not incident depression. This paper’s concept is interesting and worth. However, because the statistical analysis and the results should be revised, this paper’s evaluation is impossible to decide at this time. Also each sentence of whole manuscript is too long and including many repeats and it may be not kind to the readers who are not native like me.
(Method)
What is the definition of media contact? What kinds of communication medium? In 2011, the internet may be the most important tool in US. If this means TV news, what is the meaning of news contact before 9.11 attacks? In addition, exact media contact time of a month before 9.11 and a week after 9.11 is usually difficult to be remembered 36 months later.
(Results)
Media contact time of (one? week after 9.11) minus (one? month before 9.11) should be provided a statistical analysis in the number of depression symptom, MDD, or functional impairment. Also Bonferroni correction is needed. P is 0.00333…=0.05/15, not 0.05.
Please separate upper, middle and lower table like table 1, 2, 3. P values are better to be added in the manuscript, for readers’ easy understaning.
Middle and lower table may be ‘No’ not ‘r’.
Middle table shows the results of post disaster prevalence which include both pre and post existing depression symptom, MDD, or functional impairment and only post existing. Lower table include only post existing. If the authors would like to proof the claim, middle table should be revised to only pre + post existing data.
(Discussion)
Upper statistical analysis may change the results.
The difference of results between depression and PTSD is better to be discussed in detail.
Author Response
Comments to All Reviewers:
Thank you for the thoughtful review and for the opportunity to submit a revised version of our manuscript.
We modified the Abstract to reflect the changes in results after conducting a new analysis of the data (lines 26-33).
We simplified the text as much as possible making changes in wording (to clarify the content) and in punctuation (to enhance the flow). We did not track all of these non-substantive changes because it would be distracting to the reader. Substantive changes are identified in red font in the revised manuscript.
We added a recent publication that helps provide justification for the paper in section 1 Introduction (lines 39-57) and in section 1.2 The Current Study under Introduction (lines 74-90).
We edited and moved the original section on Clinical and Public Health Implications to Section 5 Conclusions (lines 401-411).
Response to Reviewer 3:
We addressed the concern about the statistical analysis in section 2.2 Data Analysis (lines 128-145), populated the tables with the new findings, and revised the accompanying text in section 3 Results (lines 147-194).
We appreciate the reviewer’s concern about the quality of the text. Because of concern about the length of sentences and repetition, we modified much of the text. For example, we divided some long sentences and tried to simplify the text throughout. We did not track these numerous changes because it would have made the revised manuscript difficult to read.
We addressed concerns about the measurement of media contact. As noted in section 2.1 Assessments under Materials and Methods, media contact was measured as the number of hours a day they spent “obtaining news from all sources” in the month before the 9/11 attacks and the week after the attacks (lines 120-126). Unfortunately, we did not distinguish media forms which is a limitation of the study addressed in section 4.5 Strengths and Limitations in the Discussion (lines 319-327). We appreciate the concern about recall bias and enhanced the discussion of the issue in section 4.5 Strengths and Limitations in the Discussion (lines 339-378).
To more fully address concern about recall bias, we conducted a more comprehensive review of the literature and modified our text in section 4.5 Strengths and Limitations under Discussion to reflect a more sophisticated understanding of the issue (lines 339-378). We were unable to identify any literature on memory related to media consumption and thus could not address this in the paper. We recognize that the items querying number of hours of media contact supposed more precise answers than is likely reasonable especially for pre-9/11 behavior. It might have been better to query a more subjective response but we also think that asking respondents to provide information on their pre- and post-9/11 media contact allowed them to compare their relative media use across the two time periods. We enhanced this discussion in section 4.5 Strengths and Limitations of the Discussion (lines 334-338).
The major analyses in this manuscript are cross-sectional within time periods of media contact, not across the time periods, and thus the differences in the lengths of the predisaster and postdisaster time periods for media contact are not subject to comparison. Therefore, the analysis was not adjusted for the time frames. We made a correction for multiple comparisons (10 per column) using the Bonferroni method by dividing the alpha level of significance by 10. This is noted in section 2.2 Data Analysis under Materials and Methods (lines 142-145) and in the new Tables 1 to 3 in section 3.1 Predictive Ability of Media Variables With Respect To Outcomes under Results (lines 178, 187, 194). The significant findings are fewer and we revised the text in section 3.1 Predictive Ability of Media Variables With Respect To Outcomes under Results (lines 167-194) to reflect these new findings.
As recommended, to address the different time frames for our findings in separate tables, we divided the original Table 1 into three tables, included as new Tables 1, 2, and 3 in section 3.1 Predictive Ability of Media Variables With Respect To Outcomes under Results (lines 177, 186, 193). With the revised analyses, we populated the tables with the new findings and have revised the accompanying text in section 3.1 Predictive Ability of Media Variables With Respect To Outcomes under Results (lines 167-194).
We revised the tables to include only β and p values in the columns, and thus the error identified by the reviewer’s careful attention has been removed with this revision.
We greatly appreciate the reviewer’s suggestion to change the “postdisaster” time frame to limit it to postdisaster prevalence/recurrence (Table 2), so that it will not overlap with the “incident” time frame (Table 3). We have now analyzed and presented the data for that time frame as postdisaster prevalence/recurrence findings in those without predisaster findings. The beauty of this reconceptualization and the reanalysis of this time frame is that it now presents the postdisaster findings separately for those with (Table 2, postdisaster persistence/recurrence) and without (Table 3, incidence) predisaster depressive findings. Additionally, these two tables together now provide a complete representation of the postdisaster findings across the two postdisaster subgroups with different predisaster findings. See Tables 2 and 3 (lines 186, 193)
A discussion of the distinctions between posttraumatic stress and depression is in section 4.2 Considerations Related to Disaster Trauma Exposure (lines 238-286).

Round 2
Reviewer 1 Report
The paper has been radically updated for the better. I think it's a shame no dummy table can be included, but I understand that it is no possible to do so. Furthermore, I appreciate that the analysis has been updated (wrongful use of statistics is such a major problem in science these days, especially health and psychological sciences, that it really is important to be rigorous about correct use of statistics... I won't bore you with all problems that I guess 60% of the papers in our field have with statistics, but we really need to start applying non-parametric methods when needed, instead of always thinking that linear models are applicable and even stronger than non-parametric, when that is objectively untrue). Finally, I think with the limitations correctly described, the paper is also more open about what it can and cannot say.
Author Response
Response to Reviewer 1:
Thank you for helping us improve our paper.
Reviewer 3 Report
The whole manuscript is much improved.
What is the reason of Bonferroni correction’s number? Bonferroni correction is 15 not 10, because 3 (N of depression symptoms, MDD, and functional…) * 5 (Full sample, direct, witnessed,…). None of the predisaster depression variables significantly predicted pre-9/11 news contact, and whole functional impairment may be nominally significant.
Please unify B or β in table.
Author Response
Response to Reviewer 3:
We changed our Bonferroni correction for 15 (number of columns) rather than for 10 (number of rows) as recommended, which requires that the α level of significance be set to .003 rather than .005. We adjusted the significance findings accordingly with a few small losses of significant findings.
The reanalysis revealed no differences in the key results but did eliminate significant findings for predisaster variables and for functional impairment. We have made appropriate corrections in the Abstract, Results, Tables, and Discussion to address the new findings. All changes are identified in red font. We did not track a few non-substantive changes.
Thank you for your careful attention to our paper and for helping us improve it.